

# Utilizing mutual information for detecting rare and common variants associated with a categorical trait

Leiming Sun, Chan Wang and Yue-Qing Hu

State Key Laboratory of Genetic Engineering, Institute of Biostatistics, School of Life Sciences, Fudan University, Shanghai, China

## ABSTRACT

**Background.** Genome-wide association studies have succeeded in detecting novel common variants which associate with complex diseases. As a result of the fast changes in next generation sequencing technology, a large number of sequencing data are generated, which offers great opportunities to identify rare variants that could explain a larger proportion of missing heritability. Many effective and powerful methods are proposed, although they are usually limited to continuous, dichotomous or ordinal traits. Notice that traits having nominal categorical features are commonly observed in complex diseases, especially in mental disorders, which motivates the incorporation of the characteristics of the categorical trait into association studies with rare and common variants.

**Methods.** We construct two simple and intuitive nonparametric tests, MIT and aMIT, based on mutual information for detecting association between genetic variants in a gene or region and a categorical trait. MIT and aMIT can gauge the difference among the distributions of rare and common variants across a region given every categorical trait value. If there is little association between variants and a categorical trait, MIT or aMIT approximately equals zero. The larger the difference in distributions, the greater values MIT and aMIT have. Therefore, MIT and aMIT have the potential for detecting functional variants.

**Results.** We checked the validity of proposed statistics and compared them to the existing ones through extensive simulation studies with varied combinations of the numbers of variants of rare causal, rare non-causal, common causal, and common non-causal, deleterious and protective, various minor allele frequencies and different levels of linkage disequilibrium. The results show our methods have higher statistical power than conventional ones, including the likelihood based score test, in most cases: (1) there are multiple genetic variants in a gene or region; (2) both protective and deleterious variants are present; (3) there exist rare and common variants; and (4) more than half of the variants are neutral. The proposed tests are applied to the data from Collaborative Studies on Genetics of Alcoholism, and a competent performance is exhibited therein.

**Discussion.** As a complementary to the existing methods mainly focusing on quantitative traits, this study provides the nonparametric tests MIT and aMIT for detecting variants associated with categorical trait. Furthermore, we plan to investigate the association between rare variants and multiple categorical traits.

Corresponding author
Yue-Qing Hu, yuehu@fudan.edu.cn

## INTRODUCTION

The recent outcome of genome-wide association studies (GWASs) has been successful in detecting many common single nucleotide polymorphisms (SNPs) that associate with diseases. However, just a small part of heritability can be explained by these common variants (CVs) for most diseases. This spurs more people to find rare variants (RVs) associated with complex diseases. Some studies have already shown that RVs are very important for explaining missing heritability (*Manolio et al.*, *2009*; *Zuk et al.*, *2014*). Recent advances in next generation sequencing technology offer us numerous amounts of data, which are valuable resources for RV detection and analysis. Because the frequencies of RVs are extremely low, some statistical tests for CVs are not appropriate or are powerless for RVs. New statistical methods are needed to test the association between RVs and diseases.

To this end, a lot of methods have been proposed in the past few years, and a common feature of these methods is collapsing or pooling RVs in a region to strengthen the signals. For example, the cohort allelic sum test (CAST) (*Morgenthaler & Thilly*, *2007*) collapses the genetic variants across RVs in a region to generate a new variant that indicates whether or not the subject has any RVs within the region, and CAST applies a univariate test. The Sum test (*Pan*, *2009*) generates a new variant by summing the genotype values of all the SNPs. The weighted sum statistic (WSS) (*Madsen & Browning*, *2009*) assigns various weights for every variant, which is free of models. The weighting scheme of WSS can apply to many relevant approaches. *Yi & Zhi* (*2011*) constructed a novel Bayesian generalized linear model to investigate the association of RVs and diseases. Sequence kernel association test (SKAT) (*Wu et al.*, *2011*) was developed to evaluate the association of genetic variants with a trait based on a variance component score test. Although the initial version of SKAT loses power when all genetic variants are all in the same direction of effect, the updated SKAT-O test (*Lee, Wu & Lin*, *2012*) performs well for either bidirectional or unidirectional effects. In addition, *Pan et al.* (*2014*) proposed a class of the sum of powered score $SPU(\gamma)$, which summarize the sum test and the sum of squared score test. As the power of a $SPU(\gamma)$ test depends on the selection of $\gamma$ while the best choice of $\gamma$ depends on the unexplored true relationship for RVs. Multiple SPU tests are combined by adaptive SPU (aSPU) test. Functional linear models or generalized functional linear models (GFLM) combine genetic variant positions and transform discrete genetic data to genetic variant function to detect gene respectively associated with binary or quantitative traits (*Fan et al.*, *2013*, *Fan et al.*, *2014*). Kullback–Leibler divergence based test (KLT) (*Turkmen et al.*, *2015*) detects multi-site genetic difference between cases and controls.

The above mentioned existing methods are usually applicable to the dichotomous (e.g., case-control) or continuous trait. In practice, there exist categorical traits for complex diseases, especially in mental disease studies. It is apparent that this kind of data cannot be treated as dichotomous or continuous. In this regard, the conventional likelihood based methods can be employed to detect the association between genetic variants and a categorical trait. For example, the baseline-category logit model (*Agresti*, *2012*) can be used to fit the data, and the Score test (or asymptotically equivalent Wald test or likelihood ratio

test) is convenient for the detection of associated variants since it is sufficient to calculate the maximum likelihood estimates under the null hypothesis. Although SKAT-O (*Lee, Wu & Lin*, *2012*), aSPU (*Pan et al.*, *2014*), GFLM (*Fan et al.*, *2014*) and KLT (*Turkmen et al.*, *2015*) are suitable for binary traits, the modified versions based on multiple testing corrections for all pairwise comparisons can be used to study categorical traits.

It is usually not easy to know if the baseline-category logit model could fit the observed data well. In this situation, we want to develop a nonparametric test which is free of models while having the decent power results. Mutual information, a commonly used measure in information theory, is probably a right tool to gauge the dependence between two random variables. In GWASs, some methods based on mutual information have been proposed for feature selection utilizing the genotype patterns of one or multiple variants (*Dawy et al.*, *2006*; *Brunel et al.*, *2010*). *Fan et al.* (*2011*) proposed an information gain approach based on mutual information for characterizing gene–gene and gene-environment interactions of diseases, and mutual information of two genetic variants is computed through genotype patterns. For $m$ genetic variants, there are $3^m$ probable realizations of genotype patterns. Therefore, utilizing genotype patterns is not very appropriate, especially when common variants are presented in this gene or region or the number of genetic variants is great.

We propose the mutual information based test (MIT) and adjust one (aMIT) to detect the association between multiple genetic variants and a categorical trait, which are intuitive and easily implemented. More importantly, the relative distribution of genetic variants is across all sites, not based on genotype patterns of each site. Through a range of simulation studies with varied combinations of the numbers of variants of rare causal (RC), rare non-causal, common causal (CC), and common non-causal, deleterious and protective, various minor allele frequencies (MAF) and different levels of linkage disequilibrium (LD), we demonstrate that our methods have higher power than the existing ones in terms of detecting functional variants. The robustness of the proposed test statistics to the proportions of alleles of rare to common, causal to non-causal, deleterious to protective, and linkage disequilibrium amount from weak to strong, is also shown in the simulation results. To further manifest the benefit of the proposed tests, we apply them to the data from the Collaborative Study on the Genetics of Alcoholism (COGA) which focuses on detecting ethanol-associated genes. The outputs show that MIT and aMIT are effective and powerful for dealing with the categorical trait.

## MATERIALS AND METHODS

### Notations and existing tests

Let us first introduce the notations used in this paper. Consider $n$ independent individuals and a candidate gene or region of interest harboring $m > 1$ genetic variant sites. Let $\mathbf{X}_i = (X_{i1}, \ldots, X_{im})'$ be the multi-site genotypes of the $i$th individual, where $X_{ij}$ being 0, 1, or 2 is the copy number of the minor allele at the $j$th site ($i = 1, \ldots, n, j = 1, \ldots, m$). Meanwhile, assume that there are a total of $K$ categories for a categorical trait and let $Y_i \in \{1, 2, \ldots, K\}$ denote the trait value of the $i$th individual. So the observed are $\{(\mathbf{X}_i, Y_i), i = 1, \ldots, n\}$ and the baseline-category logit model (*Agresti*, *2012*) can be employed for association study.

Let $\pi_k(\mathbf{X}_i) = P(Y_i = k|\mathbf{X}_i)$ be the conditional probability of $Y_i$ being $k$ given genotype $\mathbf{X}_i$, with $\Sigma_{k=1}^{K} \pi_k(\mathbf{X}_i) = 1$ for every $i = 1, \ldots, n$. The counts of each of the $K$ categories of $Y_i$ can be regarded as a multinomial distribution with probabilities $\{\pi_1(\mathbf{X}_i), \ldots, \pi_K(\mathbf{X}_i)\}$. We treat the last category as a baseline one (*Agresti*, *2012*) and the following models

$$\log \frac{\pi_k(\mathbf{X}_i)}{\pi_K(\mathbf{X}_i)} = \alpha_k + \mathbf{X}_i' \boldsymbol{\beta}_k, \quad k = 1, \ldots, K-1, i = 1, \ldots, n,$$

simultaneously describe the effects of $\mathbf{X}_i$ on these $K-1$ logits, where $\boldsymbol{\beta}_k = (\beta_{k1}, \ldots, \beta_{km})'$ is the corresponding vector of effect sizes of $m$ variants on the $k$th logit. The null hypothesis of no association between the genotypes and the categorical trait is $H_0 : \boldsymbol{\beta}_1 = \boldsymbol{\beta}_2 = \cdots = \boldsymbol{\beta}_{K-1} = \mathbf{0}$. In order to test if multi-site genotypes have an effect on the probabilities of falling into the different category, the Score test or asymptotically equivalent Wald test or the likelihood ratio test is applied. Compared with the Score test, the Wald or likelihood ratio test is usually computationally demanding, particular in the situation of big $m$ and $K$. Further, we realize through simulation study that the calculation of maximum likelihood estimates of the effect sizes is hard for the full model when there are a big proportion of rare variants. Therefore, we focus on the Score test in the remainder of this paper. See the Appendix for the derivation of the Score test statistic.

The asymptotic distribution of the Score test statistic under the null hypothesis is a $\chi^2$ distribution with $m(K-1)$ degrees of freedom, which implies that the power would be becoming low for big $m$ and $K$. On the other hand, the Score test depends on the model itself and in practice it is not easy for us to judge if the baseline-category logit model could fit the data properly. Therefore, the nonparametric method is appealed and expected to be effective in detecting genetic variants associated with the categorical trait. Notice that the coexistence of alleles of causal with non-causal, common with rare, deleterious with protective, is a norm in practice and should be adequately addressed in constructing such test statistic, and various levels of linkage disequilibrium should also be incorporated into study.

As SKAT-O (*Lee, Wu & Lin*, *2012*), aSPU (*Pan et al.*, *2014*), GFLM (*Fan et al.*, *2014*) and KLT (*Turkmen et al.*, *2015*) are applicable to the case of binary trait, they cannot be applied directly to the situation of categorical trait with more than two categories. As a conservative approach, we can use them to detect the associated variants between a pair of categories and then adopt Bonferroni multiple testing corrections for all possible pairwise comparisons (*Kim & Yoon*, *2011*). The corresponding tests are denoted by SKAT-O$_B$, aSPU$_B$, GFLM$_B$ and KLT$_B$, respectively. For GFLM test, we use $B$-spline to approximate genetic variant function and utilize Rao's score test which performs well as shown in *Fan et al.* (*2014*) with 10 basis functions of $B$-spline of order 4.

## Mutual information based method

Next we begin to construct the nonparametric test statistic for association study between multiple genetic variant sites and a categorical trait. For the observed categorical trait $Y = Y_1, \ldots, Y_n$ and the multi-site genotypes $\{X_{ij}, 1 \le i \le n, 1 \le j \le m\}$ of the $n$ individuals, $\sum_{i=1}^{n} X_{ij}$ is the variant frequency at site $j$, $j = 1, \ldots, m$, and $\sum_{j=1}^{m} \sum_{i=1}^{n} X_{ij}$ is the total variant frequency over the $m$ sites being considered. Let us introduce a discrete random variable $S$

to describe the distribution of "frequencies" across the region of interest as follows:

$$P(S=j) = \frac{\sum_{i=1}^{n} X_{ij} + 1}{\sum_{j=1}^{m}(\sum_{i=1}^{n} X_{ij} + 1)}, \quad j = 1, \ldots, m, \tag{1}$$

where the constant 1 is added to the counts to ensure $P(S=j) > 0$ for every $j$. Similarly, for the individuals having categorical trait value of $k$ ($k = 1, \ldots, K$), $\sum_{i=1}^{n} X_{ij} I(Y_i = k)$ and $\sum_{j=1}^{m}\sum_{i=1}^{n} X_{ij} I(Y_i = k)$ are the respective variant frequencies at every site $j$ and the overall $m$ sites, which leads to the following conditional distribution

$$P(S=j|Y=k) = \frac{\sum_{i=1}^{n} X_{ij} I(Y_i = k) + 1}{\sum_{j=1}^{m}[\sum_{i=1}^{n} X_{ij} I(Y_i = k) + 1]}, \quad j = 1, \ldots, m, \tag{2}$$

where $I(\cdot)$ is the indicator function.

Generally, if there is no association between the trait and the $m$ genetic variant sites, then the difference among the $K+1$ distributions as defined in Eqs. (1) and (2) would be small. Conversely, this kind of difference, if any, will provide us a signal of association. In order to gauge the difference, we employ the Kullback–Leibler divergence (*Kullback & Leibler*, 1951; *Turkmen et al.*, 2015) to measure the difference between two distributions. Following the idea used in the analysis of variance, we first calculate the difference between the (unconditional) distribution of $S$ and the conditional distribution of $S|Y = k$, i.e., $\text{KL}(S|Y=k, S)$, $k = 1, \ldots, K$, and then summarize these differences by the following weighted sum

$$\sum_{k=1}^{K} P(Y=k) \cdot \text{KL}(S|Y=k, S) = E_Y[\text{KL}(S|Y, S)].$$

It is easy to check

$$E_Y[\text{KL}(S|Y, S)] = \sum_{k=1}^{K} P(Y=k) \cdot \sum_{j=1}^{m} P(S=j|Y=k) \log \frac{P(S=j|Y=k)}{P(S=j)}$$

$$= \sum_{k=1}^{K}\sum_{j=1}^{m} P(S=j, Y=k) \cdot \log \frac{P(S=j, Y=k)}{P(S=j)P(Y=k)},$$

which is denoted by $\text{MI}(S, Y)$. In fact, $\text{MI}(\cdot, \cdot)$ is the mutual information, a commonly used measure in information theory to capture the amount of information in a set of variables and gauges the dependencies among them. Let MIT be the nonparametric test based on statistic $\text{MI}(S, Y)$, which is the expected divergence between the conditional distribution of $S|Y$ and unconditional distribution of $S$ with respect to $Y$.

It can be concluded that if $S$ and $Y$ are independent, then $\text{MI}(S, Y) = 0$. If $S$ depends on $Y$ weakly, then the conditional distribution of $S|Y$ is close to the unconditional distribution of $S$, which leads to a relatively small value of $\text{KL}(S|Y, S)$ and so the $\text{MI}(S, Y)$. Alternatively, a relatively large mutual information $\text{MI}(S, Y)$ could imply some dependencies between $S$ and $Y$. Therefore, it is reasonable to use MIT as a test to detect the association between the categorical trait $Y$ and $m$ genetic variant sites.

As Kullback–Leibler divergence is not symmetric, we propose the following adjusted test statistic:

$$\text{aMI}(S,Y) = \frac{1}{2}E_Y\left[\text{KL}(S|Y,S) + \text{KL}(S,S|Y)\right]$$

$$= \frac{1}{2}\left\{\sum_{k=1}^{K}\sum_{j=1}^{m}[P(S=j,Y=k) - P(S=j)P(Y=k)] \cdot \log\frac{P(S=j,Y=k)}{P(S=j)P(Y=k)}\right\},$$

and the $\text{aMI}(S,Y)$ based nonparametric test is denoted by aMIT. One feature of $\text{aMI}(S,Y)$ observed from its expression is that every summand is positive whether $P(S=j|Y=k) > P(S=j)$ or $P(S=j|Y=k) < P(S=j)$.

### Testing hypothesis

We employ the permutation strategy for evaluating the significance. Without loss of generality, we suppose that the test (which is MIT or aMIT in this paper) statistic is $T$. We first randomly shuffle the trait values of all subjects in the sample while keeping their genotypes fixed, and then apply the test statistic to the permuted data to get the corresponding test statistic $T^{(b)}$. We repeat this process for $B$ times, $b = 1,\ldots,B$. The $p$-value for the statistic $T$ is estimated as

$$p = \frac{\sum_{b=1}^{B}I(T^{(b)} \geq T)}{B}.$$

For the real data analyses of COGA, the $p$-values reported in the next section are calculated in this fashion. In the simulation study, we generate the data from a baseline-category logit model and then repeat the above procedure to obtain a $p$-value $p^{(r)}$. This process is then repeated $R$ times, that is, $r = 1, 2, \ldots, R$. Then for a fixed significance level $\alpha$, we compute

$$\sum_{r=1}^{R}I(p^{(r)} \leq \alpha)/R.$$

If the underlying model is a null case, i.e., none of the variants in the genomic region being tested is associated with the categorical trait, then this quantity is taken as the empirical type I error rate; otherwise, it is reported as the power.

## RESULTS

### Simulation study—data generation

*Simulation 1*

To evaluate the proposed methods and compare their performances with Score, SKAT-O$_B$, aSPU$_B$, GFLM$_B$ and KLT$_B$, a series of simulation studies are conducted. Specifically, we first generate a latent vector $\mathbf{Z} = (Z_1, \ldots, Z_m)'$ from a multivariate normal distribution with marginal standard normal, and covariance structure as described below, where causal variants (rare or common) and non-causal variants are randomly assigned in the $m = 24$ or 32 sites. If variants $j$ and $j'$ are both causal or both non-causal, then the correlation is set to be $\text{Corr}(Z_j, Z_{j'}) = \rho^{|j-j'|}$; otherwise the correlation is zero (*Turkmen et al., 2015*).
**Table 1  Setting of the four types of variants[a] in 10 scenarios.**

| Scenario | Number of variants | | | | Minor allele frequency | | | |
|---|---|---|---|---|---|---|---|---|
| | RC | RNC | CC | CNC | RC | RNC | CC | CNC |
| 1 | 6 | 8 | 2 | 16 | 0.005–0.01 | 0.005–0.01 | 0.1–0.3 | 0.2–0.5 |
| 2 | 6 | 16 | 2 | 8 | 0.005–0.01 | 0.01–0.05 | 0.1–0.3 | 0.1–0.3 |
| 3 | 6 | 8 | 2 | 8 | 0.005–0.01 | 0.01–0.05 | 0.1–0.3 | 0.2–0.5 |
| 4 | 6 | 8 | 2 | 8 | 0.005–0.01 | 0.005–0.01 | 0.1–0.3 | 0.1–0.3 |
| 5 | 0 | 8 | 8 | 16 | NA | 0.005–0.01 | 0.1–0.3 | 0.2–0.5 |
| 6 | 8 | 16 | 0 | 8 | 0.005–0.01 | 0.01–0.05 | NA | 0.1–0.3 |
| 7 | 0 | 8 | 8 | 16 | NA | 0.005–0.01 | 0.1–0.3 | 0.1–0.3 |
| 8 | 8 | 16 | 0 | 8 | 0.005–0.01 | 0.005–0.01 | NA | 0.1–0.3 |
| 9 | 8 | 24 | 0 | 0 | 0.005–0.01 | 0.005–0.01 | NA | NA |
| 10 | 8 | 24 | 0 | 0 | 0.005–0.01 | 0.01–0.05 | NA | NA |

**Notes.**

[a] RC, rare causal; RNC, rare non-causal; CC, common causal; CNC, common non-causal.

We take $\rho = 0, 0.5$ and $0.9$ to mimic the no, moderate and strong LD. Each $Z_j$ is then transformed to 0 (major allele) or 1 (minor allele) depending on the corresponding MAF of the variant, where MAF is selected from a uniform distribution as shown in Table 1 for different scenarios. Simulation of two $\mathbf{Z}$'s leads to a vector of genotype data denoted as $\mathbf{X}_i = (X_{i1}, \ldots, X_{im})'$, $i = 1, \ldots, n$. Note Table 1 accommodates different proportions of the numbers of variants of common to rare, causal to non-causal and varied MAFs.

The categorical trait value of the $i$th subject with genotype $\mathbf{X}_i$ is determined by $\pi_1(\mathbf{X}_i), \ldots, \pi_K(\mathbf{X}_i)$, which are deduced from the baseline-category logit model (*Agresti, 2012*) as shown in the previous section. We take $K = 3$, and $\alpha_1 = -\log(4)$ and $\alpha_2 = -\log(3)$ in the simulation study. To evaluate the type I error rates, we set $\boldsymbol{\beta}_1 = \boldsymbol{\beta}_2 = \mathbf{0}$ and the nominal significance level $\alpha = 0.05$. For power, we assign various $\boldsymbol{\beta}_1$ and $\boldsymbol{\beta}_2$ for both common and rare causal variants, as described in the following, to take different effect sizes and directions into account for investigating the influence of variants on the trait comprehensively. In scenarios 1–4 (see Table 1), the vector $\boldsymbol{\beta}_1$ of effect sizes is $(\log(3/2), \log(1/3), \log(3/2), \log(3/2), \log(1/2), \log(1/2))$ for the 6 RC variants and $(\log(11/10), \log(2/3))$ for the 2 CC variants. Accordingly, the vector $\boldsymbol{\beta}_2$ of effect sizes is $(\log(5/4), \log(1/2), \log(5/4), \log(5/4), \log(1/3), \log(1/3))$ for the 6 RC variants and $(\log(23/20), \log(1/2))$ for the 2 CC variants.

In scenarios 6 and 8–10, the vectors $\boldsymbol{\beta}_1$ and $\boldsymbol{\beta}_2$ of effect sizes for 8 RC variants are respectively

$$\boldsymbol{\beta}_1' = (\log(5/2), \log(1/3), \log(11/5), \log(11/5), \log(11/5), \log(1/2), \log(1/2), \log(1/2)),$$
$$\boldsymbol{\beta}_2' = (\log(7/5), \log(1/2), \log(2), \log(2), \log(2), \log(1/3), \log(1/3), \log(1/3)).$$

In scenarios 5 and 7, $\boldsymbol{\beta}_1$ and $\boldsymbol{\beta}_2$ for 8 CC variants are respectively

$$\boldsymbol{\beta}_1' = (\log(6/5), \log(5/6), \log(11/10), \log(11/10), \log(11/10), \log(20/23),$$
$$\log(20/23), \log(20/23)),$$
$$\boldsymbol{\beta}_2' = (\log(5/4), \log(4/5), \log(6/5), \log(6/5), \log(6/5), \log(10/13), \log(10/13), \log(10/13)).$$

Based on these parameter settings, we simulate a pool of 2,000,000 individuals with known multi-site genotypes and categorical trait. A sample of $n = 1,000$ individuals is randomly chosen from this pool in each of the $R = 1,000$ independent replications. We fix $B = 1,000$ in the permutation procedure for the calculation of $p$-value.

### Simulation 2

In this simulation set we increase the number $K$ of categories to 5 and 8 to make comparison of the test statistics. When $K = 5$, we assign $\alpha_1 = -\log(2.2), \alpha_2 = -\log(2.1), \alpha_3 = -\log(1.5)$ and $\alpha_4 = -\log(1.2)$. To evaluate the type I error rates, we set $\boldsymbol{\beta}_1 = \cdots = \boldsymbol{\beta}_4 = \mathbf{0}$ and the nominal significance level $\alpha = 0.05$. In scenarios 1–4 (see Table 1), the vectors $\boldsymbol{\beta}_1$ and $\boldsymbol{\beta}_2$ of effect sizes are the same as those in Simulation 1. The vector $\boldsymbol{\beta}_3$ of effect sizes is $(\log(27/20), \log(2/5), \log(27/20), \log(27/20), \log(2/5), \log(2/5))$ for the 6 RC variants and $(\log(23/20), \log(20/23))$ for the 2 CC variants. Accordingly, the vector $\boldsymbol{\beta}_4$ of effect sizes is $(\log(17/10), \log(10/17), \log(17/10), \log(17/10), \log(10/17), \log(10/17))$ for the 6 RC variants and $(\log(6/5), \log(5/11))$ for the 2 CC variants. In scenarios 6 and 8–10, the vectors $\boldsymbol{\beta}_1, \boldsymbol{\beta}_2, \boldsymbol{\beta}_3$ and $\boldsymbol{\beta}_4$ of effect sizes for 8 RC variants are respectively

$$\boldsymbol{\beta}_1' = (\log(3/2), \log(1/3), \log(2), \log(2), \log(2), \log(1/2), \log(1/2), \log(1/2)),$$
$$\boldsymbol{\beta}_2' = (\log(5/4), \log(1/2), \log(3), \log(3), \log(3), \log(1/3), \log(1/3), \log(1/3)),$$
$$\boldsymbol{\beta}_3' = (\log(27/20), \log(2/5), \log(5/2), \log(5/2), \log(5/2), \log(2/5), \log(2/5), \log(2/5)),$$
$$\boldsymbol{\beta}_4' = (\log(17/10), \log(10/17), \log(7/2), \log(7/2), \log(7/2), \log(2/7), \log(2/7), \log(2/7)).$$

In scenarios 5 and 7, $\boldsymbol{\beta}_1, \boldsymbol{\beta}_2, \boldsymbol{\beta}_3$ and $\boldsymbol{\beta}_4$ for 8 CC variants are respectively

$$\boldsymbol{\beta}_1' = (\log(6/5), \log(6/5), \log(23/20), \log(23/20), \log(23/20), \log(20/23), \log(20/23),$$
$$\log(20/23)),$$
$$\boldsymbol{\beta}_2' = (\log(23/20), \log(1/2), \log(28/25), \log(28/25), \log(28/25), \log(25/28), \log(25/28),$$
$$\log(25/28)),$$
$$\boldsymbol{\beta}_3' = (\log(23/20), \log(20/23), \log(23/20), \log(6/5), \log(6/5), \log(5/6), \log(5/6), \log(5/6)),$$
$$\boldsymbol{\beta}_4' = (\log(6/5), \log(5/11), \log(6/5), \log(6/5), \log(6/5), \log(10/13), \log(10/13), \log(10/13)).$$

The other parameter settings refer to Simulation 1.

When $K = 8$, we assign $\alpha_1 = -\log(2.2), \alpha_2 = -\log(2.1), \alpha_3 = -\log(1.5), \alpha_4 = -\log(1.6), \alpha_5 = -\log(2), \alpha_6 = -\log(1.8)$ and $\alpha_7 = -\log(2.1)$. To evaluate the type I error rates, we set $\boldsymbol{\beta}_1 = \cdots = \boldsymbol{\beta}_7 = \mathbf{0}$ and the nominal significance level $\alpha = 0.05$. In scenarios 1–4 (see Table 1), the vector $\boldsymbol{\beta}_1$ of effect sizes is $(\log(3), \log(1/3), \log(16/5), \log(16/5), \log(5/16), \log(5/16))$ for the 6 RC variants and $(\log(3/2), \log(2/3))$ for the 2 CC variants. Accordingly, the vector $\boldsymbol{\beta}_2$ of effect sizes is $(\log(3), \log(1/3), \log(3), \log(3), \log(1/3), \log(1/3))$ for the 6 RC variants and $(\log(31/20), \log(20/31))$ for the 2 CC variants. The vector $\boldsymbol{\beta}_3$ of effect sizes is $(\log(5/2), \log(2/5), \log(5/2), \log(5/2), \log(2/5), \log(2/5))$ for the 6 RC variants and $(\log(23/20), \log(20/23))$ for the 2 CC variants. Accordingly, the vector $\boldsymbol{\beta}_4$ of effect sizes is $(\log(17/10), \log(10/17), \log(13/10), \log(13/10), \log(10/13), \log(10/13))$ for the 6 RC variants and $(\log(6/5), \log(5/6))$ for the 2 CC variants. The vector $\boldsymbol{\beta}_5$ of effect sizes is $(\log(3), \log(1/3), \log(3), \log(3), \log(1/3), \log(1/3))$ for the 6

RC variants and $(\log(23/20),\log(20/23))$ for the 2 CC variants. The vector $\boldsymbol{\beta}_6$ of effect sizes is $(\log(27/20),\log(20/27),\log(14/5),\log(14/5),\log(5/14),\log(5/14))$ for the 6 RC variants and $(\log(6/5),\log(5/6))$ for the 2 CC variants. The vector $\boldsymbol{\beta}_7$ of effect sizes is $(\log(7/2),\log(2/7),\log(7/2),\log(7/2),\log(2/7),\log(2/7))$ for the 6 RC variants and $(\log(3/2),\log(2/3))$ for the 2 CC variants.

In scenarios 6 and 8–10, the vectors $\boldsymbol{\beta}_k(k=1,..,7)$ of effect sizes for 8 RC variants are respectively

$$\boldsymbol{\beta}'_1 = (\log(19/5),\log(5/19),\log(16/5),\log(16/5),\log(16/5),\log(5/16),$$
$$\log(5/16),\log(5/16)),$$
$$\boldsymbol{\beta}'_2 = (\log(9/2),\log(2/9),\log(3),\log(3),\log(3),\log(1/3),\log(1/3),\log(1/3)),$$
$$\boldsymbol{\beta}'_3 = (\log(4),\log(1/4),\log(5/2),\log(5/2),\log(5/2),\log(2/5),\log(2/5),\log(2/5)),$$
$$\boldsymbol{\beta}'_4 = (\log(17/10),\log(10/17),\log(13/10),\log(13/10),\log(13/10),\log(10/13),$$
$$\log(10/13),\log(10/13)),$$
$$\boldsymbol{\beta}'_5 = (\log(3),\log(1/3),\log(3),\log(3),\log(19/5),\log(5/19),\log(1/3),\log(1/3)),$$
$$\boldsymbol{\beta}'_6 = (\log(27/20),\log(2/5),\log(14/5),\log(14/5),\log(14/5),\log(5/14),$$
$$\log(5/14),\log(5/14)),$$
$$\boldsymbol{\beta}'_7 = (\log(7/2),\log(2/7),\log(7/2),\log(7/2),\log(7/2),\log(2/7),\log(2/7),\log(2/7)).$$

In scenarios 5 and 7, $\boldsymbol{\beta}_k(k=1,..,7)$ for 8 CC variants are respectively

$$\boldsymbol{\beta}'_1 = (\log(3/2),\log(2/3),\log(5/4),\log(5/4),\log(5/4),\log(4/5),\log(4/5),\log(4/5)),$$
$$\boldsymbol{\beta}'_2 = (\log(31/20),\log(20/31),\log(31/20),\log(31/20),\log(31/20),\log(20/31),\log(20/31),$$
$$\log(20/31)),$$
$$\boldsymbol{\beta}'_3 = (\log(23/20),\log(20/23),\log(23/20),\log(23/20),\log(23/20),\log(20/23),\log(20/23),$$
$$\log(20/23)),$$
$$\boldsymbol{\beta}'_4 = (\log(6/5),\log(5/6),\log(6/5),\log(6/5),\log(6/5),\log(5/6),\log(5/6),\log(5/6)),$$
$$\boldsymbol{\beta}'_5 = (\log(23/20),\log(20/23),\log(23/20),\log(23/20),\log(23/20),\log(20/23),\log(20/23),$$
$$\log(20/23)),$$
$$\boldsymbol{\beta}'_6 = (\log(6/5),\log(5/6),\log(6/5),\log(6/5),\log(6/5),\log(5/6),\log(5/6),\log(5/6)),$$
$$\boldsymbol{\beta}'_7 = (\log(3/2),\log(2/3),\log(13/10),\log(13/10),\log(13/10),\log(10/13),\log(10/13),$$
$$\log(10/13)).$$

See Simulation 1 for the other parameter settings.

### Simulation 3
Notice in the expressions of $\boldsymbol{\beta}_1$ and $\boldsymbol{\beta}_2$ in Simulation 1 that 50% of their components are positive. Now we increase this proportion to 75% and 100% and then evaluate the performance of all tests. For demonstration, we choose scenario 10 in Table 1 and set

$$\boldsymbol{\beta}'_1 = (\log(5/2),\log(3),\log(11/5),\log(11/5),\log(11/5),\log(2),\log(2),\log(2)),$$
$$\boldsymbol{\beta}'_2 = (\log(7/5),\log(2),\log(2),\log(2),\log(2),\log(3),\log(3),\log(3))$$
**Table 2** Type I error rates of seven tests with 1,000 individuals for the trait with three categories at nominal significance level 0.05.

| $\rho$ | Test | Scenario | | | | | | | | | |
|---|---|---|---|---|---|---|---|---|---|---|---|
| | | 1 | 2 | 3 | 4 | 5 | 6 | 7 | 8 | 9 | 10 |
| 0 | MIT | 0.060 | 0.050 | 0.060 | 0.052 | 0.055 | 0.040 | 0.054 | 0.068 | 0.057 | 0.052 |
| | aMIT | 0.065 | 0.040 | 0.055 | 0.053 | 0.055 | 0.040 | 0.056 | 0.067 | 0.057 | 0.046 |
| | Score | 0.060 | 0.050 | 0.040 | 0.042 | 0.060 | 0.052 | 0.040 | 0.055 | 0.058 | 0.050 |
| | SKAT-O$_B$ | 0.045 | 0.060 | 0.030 | 0.055 | 0.065 | 0.040 | 0.030 | 0.055 | 0.032 | 0.032 |
| | aSPU$_B$ | 0.057 | 0.050 | 0.045 | 0.040 | 0.060 | 0.045 | 0.044 | 0.035 | 0.057 | 0.044 |
| | GFLM$_B$ | 0.055 | 0.040 | 0.045 | 0.040 | 0.057 | 0.030 | 0.032 | 0.045 | 0.065 | 0.042 |
| | KLT$_B$ | 0.055 | 0.065 | 0.065 | 0.045 | 0.055 | 0.030 | 0.046 | 0.058 | 0.060 | 0.050 |
| 0.5 | MIT | 0.045 | 0.058 | 0.057 | 0.045 | 0.030 | 0.045 | 0.046 | 0.067 | 0.035 | 0.040 |
| | aMIT | 0.045 | 0.057 | 0.055 | 0.035 | 0.035 | 0.045 | 0.052 | 0.065 | 0.035 | 0.044 |
| | Score | 0.040 | 0.058 | 0.045 | 0.040 | 0.035 | 0.040 | 0.042 | 0.065 | 0.035 | 0.040 |
| | SKAT-O$_B$ | 0.045 | 0.058 | 0.040 | 0.035 | 0.030 | 0.050 | 0.062 | 0.050 | 0.035 | 0.040 |
| | aSPU$_B$ | 0.055 | 0.030 | 0.050 | 0.035 | 0.057 | 0.055 | 0.052 | 0.055 | 0.060 | 0.050 |
| | GFLM$_B$ | 0.035 | 0.040 | 0.055 | 0.035 | 0.045 | 0.047 | 0.054 | 0.055 | 0.035 | 0.040 |
| | KLT$_B$ | 0.065 | 0.045 | 0.065 | 0.035 | 0.045 | 0.047 | 0.048 | 0.051 | 0.060 | 0.048 |
| 0.9 | MIT | 0.057 | 0.043 | 0.055 | 0.050 | 0.050 | 0.057 | 0.050 | 0.055 | 0.045 | 0.048 |
| | aMIT | 0.059 | 0.043 | 0.060 | 0.045 | 0.060 | 0.057 | 0.050 | 0.055 | 0.045 | 0.048 |
| | Score | 0.057 | 0.030 | 0.047 | 0.055 | 0.040 | 0.065 | 0.040 | 0.045 | 0.035 | 0.036 |
| | SKAT-O$_B$ | 0.035 | 0.045 | 0.030 | 0.035 | 0.057 | 0.040 | 0.034 | 0.060 | 0.038 | 0.038 |
| | aSPU$_B$ | 0.057 | 0.035 | 0.065 | 0.055 | 0.045 | 0.035 | 0.060 | 0.045 | 0.045 | 0.034 |
| | GFLM$_B$ | 0.045 | 0.035 | 0.045 | 0.040 | 0.035 | 0.045 | 0.040 | 0.055 | 0.040 | 0.046 |
| | KLT$_B$ | 0.058 | 0.055 | 0.065 | 0.058 | 0.060 | 0.045 | 0.058 | 0.045 | 0.050 | 0.046 |

in the case of 100% proportion and

$$\boldsymbol{\beta}_1' = (\log(5/2), \log(3), \log(11/5), \log(11/5), \log(11/5), \log(2), \log(1/2), \log(1/2)),$$
$$\boldsymbol{\beta}_2' = (\log(7/5), \log(2), \log(2), \log(2), \log(2), \log(3), \log(1/3), \log(1/3))$$

in case of 75%. All the other parameter settings remain the same as Simulation 1.

## Simulation results
### Type I error rates
To check the validity of our proposed tests MIT and aMIT, in addition to Score, SKAT-O$_B$, aSPU$_B$, GFLM$_B$ and KLT$_B$, we begin by presenting the type I error rates for each test with the null simulation parameters under different magnitude of LD. The empirical type I error rates of all tests of Simulation 1 and 2 are respectively summarized in Table 2, Table S1 ($K = 5$) and Table S2 ($K = 8$). These tables demonstrate that the estimated type I error rates of all tests are not significantly different from the predetermined nominal level for different LD amounts. As expected, SKAT-O$_B$, aSPU$_B$, GFLM$_B$ and KLT$_B$ were a little bit less conservative, as can be seen in Table 2, Table S1 ($K = 5$) and Table S2 ($K = 8$).

### Power comparison
The following results are for the evaluation of the power of the proposed tests MIT and aMIT as well as Score, SKAT-O$_B$, aSPU$_B$, GFLM$_B$ and KLT$_B$ in different scenarios. Note that
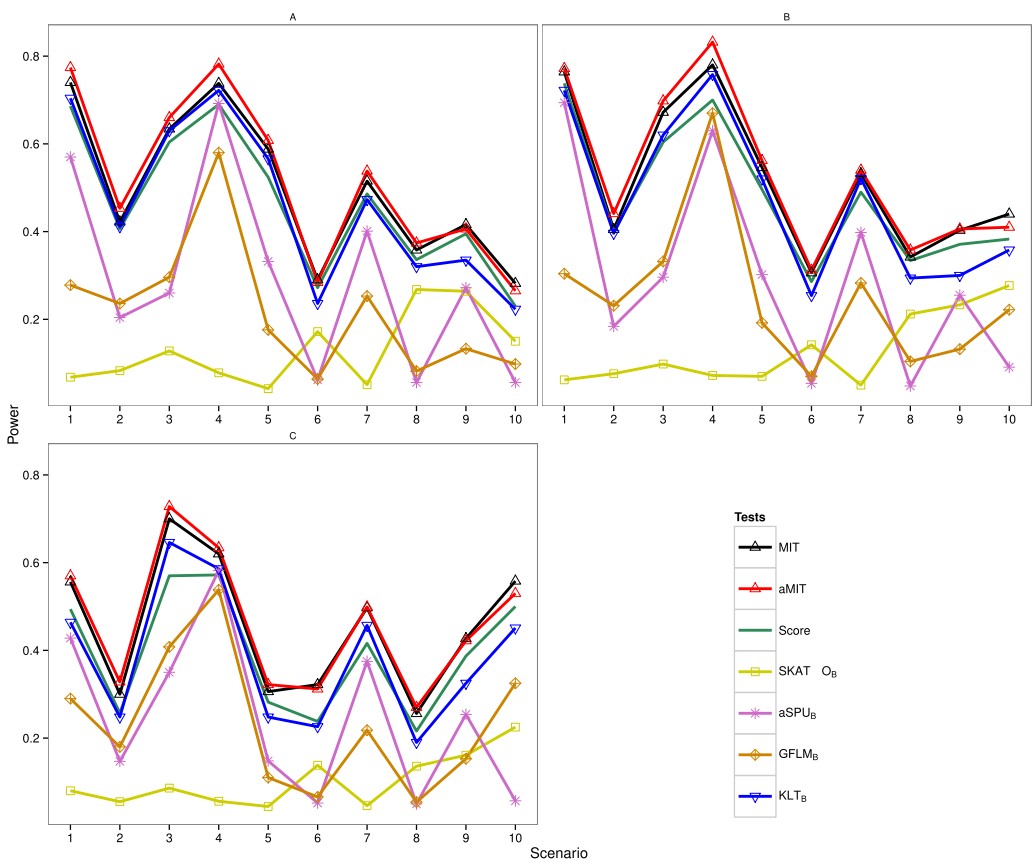

**Figure 1** **Power results of seven tests with 1,000 individuals for a trait with three categories.** (A) $\rho = 0$; (B) $\rho = 0.5$; (C) $\rho = 0.9$.

SKAT-O, aSPU and GFLM are implemented by using R packages. The power comparison of these seven tests in Simulation 1 is summarized in Fig. 1. It can be seen from this figure that when $\rho$ increases from 0.5 to 0.9, the power results of MIT and aMIT decrease obviously in scenarios 1–4. This illustrates that the discrepancies between distributions of genetic variants given each categorical trait value decrease when the variants are in strong LD structure. In addition, power results of MIT and aMIT are relatively low in scenarios 6, 8, 9 and 10 with only RC variants. According to Fig. 1, aMIT has higher power than MIT for varied $\rho$ in almost all scenarios. In strong LD structure, the advantage of aMIT becomes more obvious in scenarios 1–4. We could conclude that aMIT is more appropriate when both common and rare causal variants are present in the region of study.

For the four conservative tests SKAT-O$_B$, aSPU$_B$, GFLM$_B$ and KLT$_B$, they are not the winners for all 10 scenarios and 3 LD structures as expected. In scenarios 6 and 8, the performance of SKAT-O$_B$ is better than aSPU$_B$ or GFLM$_B$. aSPU$_B$ or GFLM$_B$ has attracting performance in scenarios 1–5 and 7. The results of power comparison of SKAT$_B$ and GFLM$_B$ are consistent with those in *Fan et al.* (*2014*). KLT$_B$ is always the winner among the four conservative tests. It is not surprising to see that Score test has higher power than KLT$_B$ in some scenarios as both the data and Score test statistic are from

**Table 3  Power results of seven tests for 75% and 100% proportions of positive causal variants in causal ones with 1,000 individuals for a trait with three categories at nominal significance level 0.05 in scenario 10.**

| Proportion | $\rho$ | MIT | aMIT | Score | SKAT-O$_B$ | aSPU$_B$ | GFLM$_B$ | KLT$_B$ |
|---|---|---|---|---|---|---|---|---|
| 75% | 0 | 0.519 | 0.491 | 0.513 | 0.317 | 0.070 | 0.198 | 0.415 |
| | 0.5 | 0.549 | 0.519 | 0.526 | 0.317 | 0.069 | 0.182 | 0.398 |
| | 0.9 | 0.653 | 0.634 | 0.590 | 0.307 | 0.066 | 0.296 | 0.520 |
| 100% | 0 | 0.601 | 0.56 | 0.639 | 0.540 | 0.123 | 0.517 | 0.506 |
| | 0.5 | 0.742 | 0.712 | 0.736 | 0.656 | 0.143 | 0.649 | 0.672 |
| | 0.9 | 0.969 | 0.959 | 0.894 | 0.853 | 0.210 | 0.924 | 0.946 |

the same baseline-category logit model, but it has lower power than KLT$_B$ in the other scenarios. With the exception of scenario 7 with $\rho = 0.5$, either MIT or aMIT is the winner among those seven tests. On average, aMIT is superior to MIT in Simulation 1.

The results of Simulation 2 are shown in Fig. S1 ($K = 5$) and Fig. S2 ($K = 8$), and MIT and aMIT tests were almost always the winners. Because the degrees of freedom of the Score test statistic are respectively $4m$ and $7m$ ($m = 24$ or $32$), the gap between Score and the winner increases when the number of categories increases. Among the four conservative tests, KLT$_B$ performs best when $\rho = 0$ except scenario 9, but does not have high power in scenarios 4 and 9 with $\rho = 0.5$ and in most scenarios with $\rho = 0.9$. SKAT-O$_B$ performs better than aSPU$_B$ and GFLM$_B$ in scenarios 6, 8 and 10. In Fig. S2, the gap between Score and the winner is noticeable in scenario 9 with $\rho = 0.9$. When $\rho = 0$ and 0.5, KLT$_B$ is the winner among the four conservative tests except scenario 2. aSPU$_B$ and GFLM$_B$ perform better than SKAT-O$_B$ in scenarios 5 and 7.

The results of Simulation 3 are shown in Table 3. The Simulation 3 focuses on 75% and 100% proportions of positive causal variants with 1,000 individuals. The power of all tests increases with the LD amounts. MIT or aMIT performs better than Score with $\rho = 0.5$ and 0.9 in both 75% and 100% cases, but Score is the winner in situation of linkage equilibrium. MIT or aMIT is recommended for the situations in which multiple sites in a region are in, weak or strong, LD.

It is demonstrated for the most situations in Fig. 1, Figs. S1, S2 and Table 3 that the performance of MIT or aMIT is superior to the existing methods in term of detecting variants. For scenarios shown in Fig. 1, Figs. S1 and S2, aMIT performs better than MIT, especially for big LD amounts or they have approximately equal power. When the proportion of positive causal variants increases as shown in Table 3, MIT has some advantages. In summary, MIT and aMIT have superior performance in a range of scenarios, compared to several existing tests.

## Application to data from collaborative study on the genetics of alcoholism study

This study was designed to locate the specific genes influencing a person's likelihood of developing alcoholism. There are a total of 1,945 subjects in the COGA data which contain 1,594 ones in Health Research group and 351 ones in Alcoholism and Related Conditions

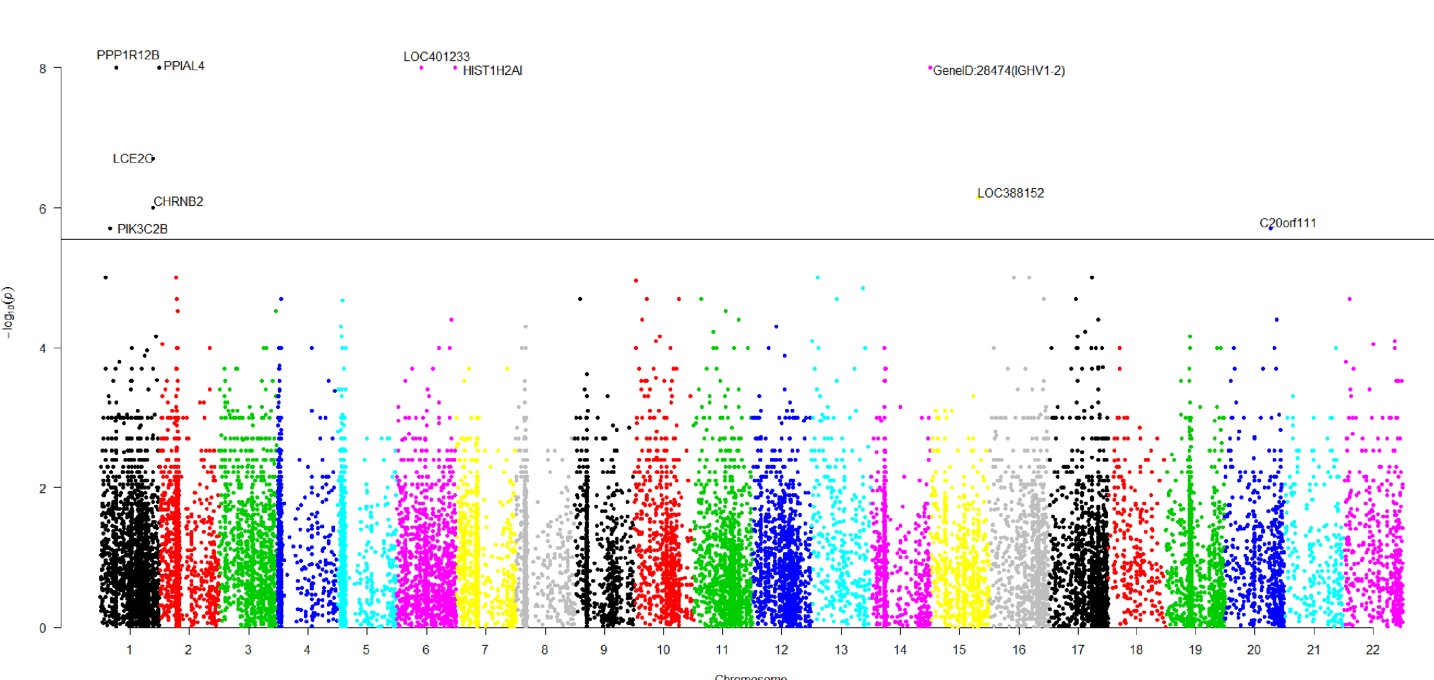

**Figure 2** **The manhattan plot of all *p*-values of genes based on MIT for COGA data.** Horizontal black line represents the threshold of $2.798\times10^{-6}$.

group. All subjects were tested through the Semi-Structured Assessment for the Genetics in Alcoholism (*Bucholz et al.*, *1994*).

Genotyping was sequenced by the Illumina Human 1M DNA Analysis at the Center for Inherited Disease Research. The sequence data contain 1,072,820 SNPs in 18,946 genes on all chromosomes. There are 32 subjects for which phenotypes are released, but no genotype data is available, so they are removed in the subsequent analyses. After removing variants that have no sequence variation (all homozygous for the common allele), there are 17,868 genes on autosomes with 1,913 subjects.

ALDX1 (Alcohol diagnosis—DSM3R (*American Psychiatric Association*, *1994*) and *Feighner et al.* (*1972*) is a categorical trait with three categories (1-pure unaffected, 3-unaffected with some symptoms and 5-affected) in COGA data, which depicts the severity of alcohol dependence. The data used for analyzing are collected from dbGaP at http://www.ncbi.nlm.nih.gov/projects/gap/cgi-bin/study.cgi?study_id=phs000125.v1.p1 through dbGaP accession number phs000125.v1.p1. A number of researchers involve in this study and they analyze SNP one by one or one whole chromosome or utilize other related traits (*Edenberg et al.*, *2010*; *Zhang, Liu & Wang*, *2010*; *Wang et al.*, *2011*). We utilize ALDX1 to reanalyze COGA data and detect genes associated with ALDX1.

### A genome-wide scan

Using Bonferroni correction of multiple testings, a genome-wide significance of 0.05 requires the *p*-value based on gene to be less than $0.05/17,868 = 2.798\times10^{-6}$. Figure 2 is the Manhattan plot of all *p*-values of genes based on MIT. The results for aMIT are similar

and we have skipped them here. Due to the time limitation in the permutation procedure, the $p$-values of less than $10^{-8}$ will be denoted by $-\log(10^{-8}) = 8$ in Fig. 2. It is concluded from Fig. 2 that there are a total of 10 significant genes.

We are interested in the functions of these significant genes and their relationships with alcohol dependence. Gene PPP1R12B, also known as MYPT2, is a subunit of protein phosphatase 1 (PP1) and is mainly reflected in the heart, skeletal muscle and the brain (*Pham et al., 2012*). There is no doubt that the problem of binge drinking, particularly by young, needs to be addressed urgently, to prevent cognitive impairment, which could lead to irreversible brain damage (*Ward, Lallemand & De witte, 2009*). In the neural development stage, different ethanol treatments can change the expression of gene PPP1R12B in the adult brain (*Kleiber et al., 2013*). PP1 is in the alcoholism pathway for human; see Figs. S3 from the Kyoto Encyclopedia of Genes and Genomes (KEGG) (http://www.genome.jp/dbget-bin/www_bget?map05034). N-methyl-D-aspartate (NMDA) receptor which is a glutamate receptor and ion channel protein can be dephosphorylated by PP1. DARPP32 (dopamine- and cAMP-regulated phosphoprotein of 32 kDa) is a protein expressed mainly in neurons and restrains PP1. DARPP32 augments NMDA receptor phosphorylation, which adds channel function and offsets acute prohibition about ethanol (*Spanagel, 2009*). PP1 plays the role of a bridge between NMDA and DARPP32. The relationship between alcohol, dopamine and glutamate may be likely to develop "binge drinking" (alcoholism) (*Ward, Lallemand & De witte, 2009*).

Gene HIST1H2AI is one of the histone H2A family. H2A is also in the alcoholism pathway (see Figs. S3). The position of Histones is very important in transcription regulation, the repair and recombination of DNA and the stability of chromosome. A complex set of histone repair regulate DNA, which are also known as histone codes, and the rebuilding of nucleosome (http://www.genecards.org/cgi-bin/carddisp.pl?gene=HIST1H2AI&keywords=HIST1H2AI). The histone octamer contains two molecules of each of the histones H2A, H2B, H3 and H4, around which the DNA wraps (*Mandrekar, 2011*). For alcoholism, the roles of Histone acetylation and methylation are very important in brain and peripheral tissues (*Starkman, Sakharkar & Pandey, 2012*).

The brain could be affected by alcoholism which results in tolerance and dependence. Moreover alcoholism will have a serious harmful effects in other organs. PPIAL4A is a protein-coding gene and associates with liver cancer (http://www.genecards.org/cgi-bin/carddisp.pl?gene=PPIAL4A&keywords=PPIAL4A).

In addition, nicotine use is closely related with ethanol intake. Smokers will be prone to drink more ethanol than the peers of nonsmokers (*Britt & Bonci, 2013*). Drinkers are very liable to smoke; moreover, alcoholics are often dependent on tobacco (*Drobes, 2002*). Gene CHRNB2 is in nicotine addiction pathway. Moreover, CHRNB2 associates with tobacco- and ethanol-associated traits, and markers mediating early responses to nicotine and alcohol can be found in CHRNB2 (*Ehringer et al., 2007*). C20orf111 resides in marker D20S119, which has a high LOD score using the alcoholism phenotype (*Hill et al., 2004*).

The function of gene LOC401233 is similar to that of gene HTATSF1P2, which associates with human immunodeficiency virus (HIV-1). Both alcohol abuse and HIV infection are

believed to compromise immune function and the nervous system (*Silverstein & Kumar*, *2014*). In fact, the progression of HIV may be aggravated by alcohol abuse and HIV patients are more likely to use alcohol than the general subjects (*Pandrea et al.*, *2010*).

### Functional genes

For comparison of all tests, we focus on functional genes as shown in literatures. There are a total of 304 genes associated with alcohol dependence based on National Center for Biotechnology Information (http://www.ncbi.nlm.nih.gov/). From them, we select 202 functional genes which are for human sapiens and are present in COGA data. Top 10 genes detected by aMIT are shown in Table 4. It is apparent that MIT or aMIT performs much better than other tests. Note for the MAF shown in the third column in Table 4 that there are both rare and common variants in these genes. Meanwhile, Figure 3 depicts a five-way Venn diagram illustrating either the concordance or discordance of genes found to be significant with *p*-values less than 0.05 for five tests. The counts shown where all ellipses overlap indicate the number of (concordant) genes which are detected by all tests, while the counts that are in only one ellipse denote the number of (discordant) genes that are detected by the corresponding test. The counts of each ellipse for aMIT, Score, SKAT-O$_B$, aSPU$_B$, GFLM$_B$ and KLT$_B$ tests are respectively 90, 21, 46, 52, 34 and 68. As the Score test only detects 21 functional genes and is excluded in Venn diagram. There are six genes detected by all five tests in Fig. 3. It is concluded from this figure that aMIT test is the most powerful tool to detect functional genes.

Alcohol dependence can affect our brain, heart, immune system, and even our life. Long-term use of alcohol can cause serious health complications, affecting almost most of the organs in our body. The MIT or aMIT tests perform better than other tests for detecting the association between function genes and categorical traits.

## DISCUSSION

In the era of GWASs, tremendous efforts are devoted to developing methods for investigating genetic variants associated with quantitative/binary/ordinal traits. It seems true that the categorical trait receives little attention. One possible reason for this is the lack of appropriate/powerful statistical method. Although the conventional baseline-category logit model could be used to explore the relationship between multiple variants and categorical trait, it would suffer from low power when the number of variants/categories increases, or in some situations we do not have sufficient reasons to use this model to fit the observed data. It is desirable to develop nonparametric test with high power to deal with categorical traits.

Mutual information, a dependence measure used frequently in information theory, can be used to gauge the strength of relatedness between multi-site genotypes and a categorical trait. It is actually a weighted sum of Kullback–Leibler divergence. Specifically, given every categorical trait value, we can calculate the conditional distribution of variants across a region, and then compare it to the overall distribution. The mutual information based test statistic is nonparametric, free of mode of inheritance, intuitive and easily implemented.

Sun et al. (2016), *PeerJ*, DOI 10.7717/peerj.2139

**Table 4** The top 10 genes detected by aMIT in 202 functional genes.

| Gene | Position | Number of Variants | MAF | MIT | aMIT | Score | SKAT-O$_B$ | aSPU$_B$ | GFLM$_B$ | KLT$_B$ |
|---|---|---|---|---|---|---|---|---|---|---|
| CHRNB2 | 1q21.3 | 5 | 0.0680–0.2700 | 1.00E–06 | 2.00E–06 | 3.00E–04 | 6.00E–05 | 0.0030 | 0.0048 | 0.0120 |
| SERINC2 | 1p35.1 | 35 | 0.0010–0.4864 | 1.00E–05 | 3.00E–05 | 0.4055 | 0.1002 | 0.0012 | 0.0599 | 3.00E–04 |
| GABRG2 | 5q34 | 223 | 0.0018–0.4997 | 5.00E–05 | 5.00E–05 | 0.6350 | 0.0671 | 9.00E–04 | 0.0103 | 1.20E–04 |
| ADH5 | 4q23 | 46 | 0.0003–0.4927 | 4.14E–04 | 6.70E–05 | 0.1297 | 3.00E–04 | 0.1617 | 0.0259 | 3.00E–04 |
| PKNOX2 | 11q24.2 | 101 | 0.0042–0.4979 | 1.00E–04 | 1.00E–04 | 0.5886 | 0.1710 | 0.0042 | 0.1151 | 0.0150 |
| OPA3 | 19q13.32 | 31 | 0.0031–0.4579 | 1.00E–04 | 1.00E–04 | 0.0163 | 9.00E–04 | 3.00E–04 | 0.0033 | 0.3471 |
| GAD2 | 10p11.23 | 71 | 0.0003–0.4937 | 2.00E–04 | 1.00E–04 | 0.5301 | 0.0119 | 0.0024 | 0.2421 | 0.0030 |
| GABRR2 | 6q15 | 49 | 0.0340–0.4830 | 1.00E–04 | 1.00E–04 | 8.94E–04 | 0.0031 | 0.0150 | 0.0607 | 0.0030 |
| LARGE | 22q12.3 | 596 | 0.0013–0.4992 | 1.60E–04 | 1.50E–04 | 0.6226 | 0.0036 | 0.0120 | 0.2751 | 6.00E–04 |
| NXPH2 | 2q22.1 | 276 | 0.0037–0.4997 | 3.00E–04 | 3.00E–04 | 0.1139 | 0.0150 | 0.0330 | 0.3196 | 0.0234 |

**Notes.**

MAF, minor allele frequency.

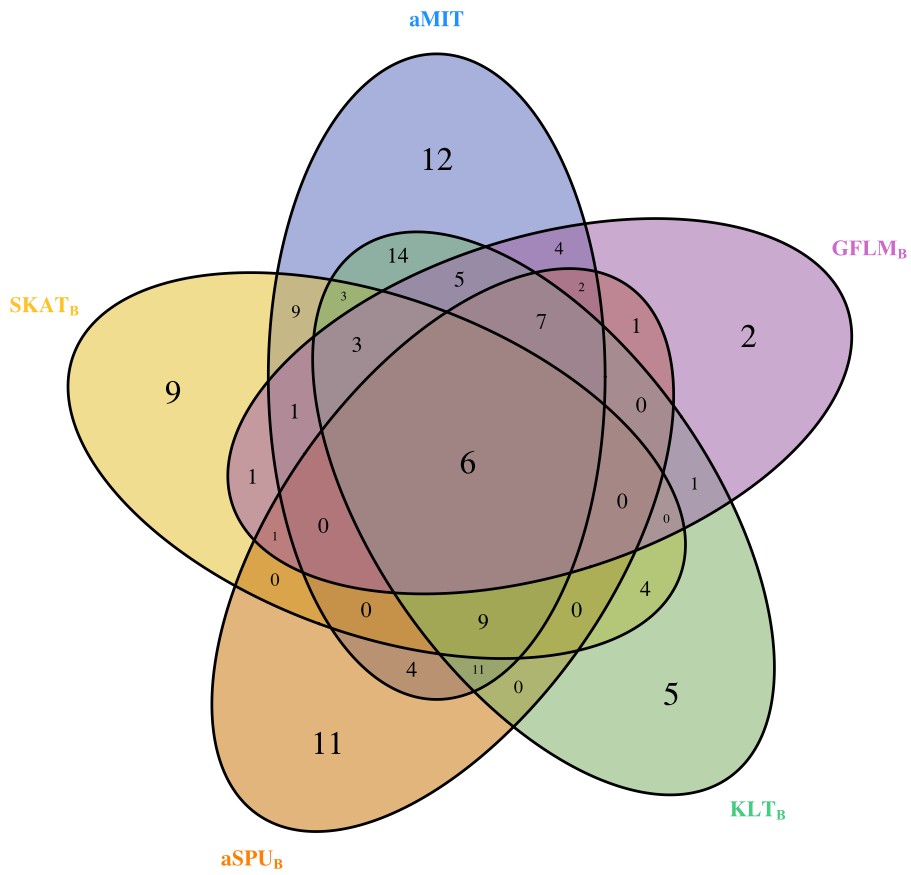

**Figure 3** The Venn diagram of the counts of genes found significant with *p*-value less than 0.05 for five tests from 202 funtional genes.

It is a common sense that in a genetic region there exist both common and rare variants, causal and non-causal variants, deleterious and protective variants, mild and strong linkage disequilibrium, which are adequately addressed for the proposed test statistics. In this paper, all these issues along with various MAFs are taken into account in the set-ups of simulation. The surprising performance of the proposed test statistics is demonstrated in the simulation results. Meanwhile, we realize that our proposed tests are not applicable to the special case of $m = 1$ and their performances are not good when all variants are causal and have the same effect sizes since there are not apparent differences among the $K + 1$ distributions as defined in Eqs. (1) and (2) . Fortunately, a gene/region harboring multiple genetic variants with varied MAFs, effect sizes, effect directions is a norm. In the case of a small number of variant sites, the genotype pattern approach is an alternative (*Dawy et al.*, *2006*; *Brunel et al.*, *2010*). Further, their efficiency and feasibility are illustrated in the analyses of real data from COGA. The association study between genes and multiple categorical traits is for our future research. The R package of MIT and aMIT is provided in the Supplemental Information files.

## CONCLUSION

In this paper, we develop MIT and aMIT, two nonparametric tests which can effectively detect the association between genetic variants and categorical traits. One feature of MIT or aMIT is that they are free of models. Observed from simulation studies and real data analyses, the proposed tests perform better than the conventional Score test. Therefore, MIT and aMIT are recommended for identifying genetic variants associated with categorial traits.

## ACKNOWLEDGEMENTS

The authors would like to thank the Academic Editor and anonymous reviewers for their insightful and constructive comments, which lead to an improved manuscript.

## APPENDIX. SCORE TEST STATISTIC

In this appendix, we sketch the derivation of score test statistic for testing $H_0 : \boldsymbol{\beta}_1 = \boldsymbol{\beta}_2 = \cdots = \boldsymbol{\beta}_{K-1} = \mathbf{0}$ based on the following baseline-category logit model:

$$\log \frac{\pi_k(\mathbf{X}_i)}{\pi_K(\mathbf{X}_i)} = \alpha_k + \mathbf{X}_i'\boldsymbol{\beta}_k, \qquad k = 1, \ldots, K-1, i = 1, \ldots, n,$$

which implies $\pi_K(\mathbf{X}_i) = [1 + \sum_{k=1}^{K-1} \exp(\alpha_k + \mathbf{X}_i'\boldsymbol{\beta}_k)]^{-1}$ for every $i$. For subject $i$, let $(y_{i1}, \ldots, y_{iK})'$ represent his/her multinomial trait, where $y_{ik} = 1$ when the trait is in category $k$, and $y_{ik} = 0$ otherwise, so $\sum_{k=1}^{K} y_{ik} = 1$. Since $\pi_K = 1 - (\pi_1 + \cdots + \pi_{K-1})$ and $y_{iK} = 1 - (y_{i1} + \cdots + y_{i,K-1})$, the log likelihood function (*Agresti, 2012*) is

$$l = \log \prod_{i=1}^{n} \left[ \prod_{k=1}^{K} \pi_k(\mathbf{X}_i)^{y_{ik}} \right] = \sum_{i=1}^{n} \left[ \sum_{k=1}^{K-1} y_{ik} \log \pi_k(\mathbf{X}_i) + (1 - \sum_{k=1}^{K-1} y_{ik}) \log \pi_K(\mathbf{X}_i) \right]$$

$$= \sum_{i=1}^{n} \left[ \sum_{k=1}^{K-1} y_{ik} \log \frac{\pi_k(\mathbf{X_i})}{\pi_K(\mathbf{X}_i)} + \log \pi_K(\mathbf{X}_i) \right]$$

$$= \sum_{i=1}^{n} \sum_{k=1}^{K-1} y_{ik}(\alpha_k + \mathbf{X}_i'\boldsymbol{\beta}_k) - \sum_{i=1}^{n} \log(1 + \sum_{k=1}^{K-1} \exp(\alpha_k + \mathbf{X}_i'\boldsymbol{\beta}_k)).$$

The first partial derivatives of the log likelihood with respect to parameters $\alpha_k$ and $\boldsymbol{\beta}_k$ ($1 \le k \le K-1$) are respectively

$$\frac{\partial l}{\partial \alpha_k} = \sum_{i=1}^{n} y_{ik} - \sum_{i=1}^{n} \pi_K(\mathbf{X}_i) \exp(\alpha_k + \mathbf{X}_i'\boldsymbol{\beta}_k),$$

$$\frac{\partial l}{\partial \boldsymbol{\beta}_k} = \sum_{i=1}^{n} y_{ik} \mathbf{X}_i - \sum_{i=1}^{n} \pi_K(\mathbf{X}_i) \exp(\alpha_k + \mathbf{X}_i'\boldsymbol{\beta}_k) \mathbf{X}_i.$$

Under $H_0$, the maximum likelihood estimators $\widehat{\alpha}_k$ of unknown parameters $\alpha_k$ ($1 \le k \le K-1$) can be solved from the following $K-1$ equations

$$\left. \frac{\partial l}{\partial \alpha_k} \right|_{H_0} = 0, \qquad k = 1, \ldots, K-1,$$

which can be simplified to $[1 + \sum_{k=1}^{K-1} \exp(\alpha_k)]^{-1} \exp(\alpha_k) = \bar{y}_{\cdot k}$, where $\bar{y}_{\cdot k} = \frac{1}{n} \mathbf{1}' \mathbf{y}_k$, $\mathbf{1}$ is the all 1 vector of length $n$, and $\mathbf{y}_k = (y_{1k}, y_{2k}, \ldots, y_{nk})'$, $k = 1, \ldots, K-1$.

The score vector $\mathbf{U} = (\mathbf{U}'_1, \ldots, \mathbf{U}'_{K-1})'$, where $\mathbf{U}_k$ is $\partial l / \partial \boldsymbol{\beta}_k$ evaluated at $H_0$ and $\alpha_k = \widehat{\alpha}_k$, $k = 1, \ldots, K-1$. Based on the expression of $\partial l / \partial \boldsymbol{\beta}_k$, we have

$$\mathbf{U}_k = \sum_{i=1}^{n} (y_{ik} - \bar{y}_{\cdot k}) \mathbf{X}_i = \mathbf{X}'(\mathbf{I}_n - \frac{1}{n} \mathbf{1}\mathbf{1}') \mathbf{y}_k,$$

where $\mathbf{X}' = (\mathbf{X}_1, \ldots, \mathbf{X}_n)$ and $\mathbf{I}_n$ is the $n \times n$ identity matrix. Note that when $H_0$ holds and $\alpha_k = \widehat{\alpha}_k$, we have $Cov(\mathbf{y}_k, \mathbf{y}_k) = \bar{y}_{\cdot k}(1 - \bar{y}_{\cdot k})\mathbf{I}_n$ and $Cov(\mathbf{y}_k, \mathbf{y}_{k'}) = -\bar{y}_{\cdot k}\bar{y}_{\cdot k'}\mathbf{I}_n$, for $1 \le k \ne k' \le K-1$. So the corresponding variance–covariance matrix $\mathbf{V}$ of $\mathbf{U}$ is

$$\begin{bmatrix} \bar{y}_{\cdot 1}(1 - \bar{y}_{\cdot 1}) & -\bar{y}_{\cdot 1}\bar{y}_{\cdot 2} & \cdots & -\bar{y}_{\cdot 1}\bar{y}_{\cdot K-1} \\ -\bar{y}_{\cdot 2}\bar{y}_{\cdot 1} & \bar{y}_{\cdot 2}(1 - \bar{y}_{\cdot 2}) & \cdots & -\bar{y}_{\cdot 2}\bar{y}_{\cdot K-1} \\ \vdots & \vdots & \vdots & \vdots \\ -\bar{y}_{\cdot K-1}\bar{y}_1 & -\bar{y}_{\cdot K-1}\bar{y}_{\cdot 2} & \cdots & \bar{y}_{\cdot K-1}(1 - \bar{y}_{\cdot K-1}) \end{bmatrix} \otimes \mathbf{X}'(\mathbf{I}_n - \frac{1}{n}\mathbf{1}\mathbf{1}')\mathbf{X},$$

where $\otimes$ is the matrix Kronecker product. The score test statistic is $\mathbf{U}'\mathbf{V}^{-1}\mathbf{U}$ which has an asymptotic $\chi^2$ distribution with $rank(\mathbf{V})$ degrees of freedom.

### Funding

This research was supported in part by the National Natural Science Foundation of China (11571082, 11171075), National Basic Research Program of China (2012CB316505), and the Scientific Research Foundation of Fudan University. The funders had no role in study design, data collection and analysis, decision to publish, or preparation of the manuscript.

### Grant Disclosures

The following grant information was disclosed by the authors:
National Natural Science Foundation of China: 11571082, 11171075.
National Basic Research Program of China: 2012CB316505.
Scientific Research Foundation of Fudan University.

### Competing Interests

The authors declare there are no competing interests.

### Author Contributions

- Leiming Sun conceived and designed the experiments, performed the experiments, analyzed the data, contributed reagents/materials/analysis tools, wrote the paper, prepared figures and/or tables.
- Chan Wang conceived and designed the experiments, performed the experiments, contributed reagents/materials/analysis tools.

- Yue-Qing Hu conceived and designed the experiments, performed the experiments, analyzed the data, wrote the paper, prepared figures and/or tables, reviewed drafts of the paper.

## Data Availability

CIDR: Collaborative Study on the Genetics of Alcoholism Case Control Study; dbGaP Study Accession: phs000125.v1.p1; http://www.ncbi.nlm.nih.gov/projects/gap/cgi-bin/study.cgi?study_id=phs000125.v1.p1.

## Supplemental Information

Supplemental information for this article can be found online at http://dx.doi.org/10.7717/peerj.2139#supplemental-information.

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
