# Peer review of "Utilizing mutual information for detecting rare and common variants associated with a categorical trait"

_PeerJ, doi:10.7717/peerj.2139_

## Round 0.1 · original submission · Major Revisions

The manuscript has been carefully revised by two external reviewers. Overall, we think that the manuscript has merit but it is not suitable for publication as it currently stands. Indeed, the reviewers raised major criticisms that required attention and to perform new calculations and analyses. We would be glad to evaluate a major revised version of the manuscript where all the comments raised by the reviewers will be properly addressed.

Reviewer 1 ·

Basic reporting

I have read the paper by Sun et al. The authors utilized mutual information for gene-based association analysis of categorical trait of complex disorders. The paper is well-written and the overall explanation is clear. My main concerns are that there are different methods to perform gene-based analysis, but the authors only compare their methods with mixed model based tests, i.e., SKAT and SKAT-O and SPU. I suggest that the authors add comparison with tests of functional regression models. The references can be found below:

1. Fan et al. (2013) Functional linear models for association analysis of quantitative traits. Genetic Epidemiology 37:726-742.
2. Fan et al. (2014) Generalized functional linear models for case-control association studies. Genetic Epidemiology 38:622-637.

The R codes can be found in

http://www.nichd.nih.gov/about/org/diphr/bbb/software/fan/Pages/default.aspx

If the authors are willing to make the changes, I can review the paper again.

Experimental design

ok

Validity of the findings

see my report

Reviewer 2 ·

Basic reporting

The manuscript introduces two tests based on mutual information and permutation testing for association analysis of common and rare variants with categorical traits.

Overall the writing is clear enough to be understandable, but it contains a rather large number of typos/grammatical errors and would need to be language-edited before publication.

The manuscript lists a number of existing tests for association analysis, but it does not reference any mutual information based approaches. However, using mutual information in association testing together with permutation based null distribution is not a novel idea in general and the manuscript does not discuss its relationship to existing MI approaches at all, making it difficult to know what the real contribution over existing literature is. For example, Section 2.3 of
Bruned et al. MISS: a non-linear methodology based on mutual information for genetic association studies in both population and sib-pairs analysis, Bioinformatics, (2010) 26 (15): 1811-1818. https://bioinformatics.oxfordjournals.org/content/26/15/1811.full
describes essentially the same approach for association testing (albeit not in categorical traits). As another example, the following seems relevant:
Dawy et al, Gene mapping and marker clustering using Shannon's mutual information, IEEE/ACM Transactions on Computational Biology and Bioinformatics (Volume: 3, Issue: 1), http://dx.doi.org/10.1109/TCBB.2006.9

A better literature review would be required for publication.

In constructing the MI based test, the definition P(S = j) is not motivated clearly, and thus intuition behind the whole test is not very clear. For example, if the studied site would contain only one variant (that is, m = 1), P(S = 1) = 1 always, so it seems that the test is not applicable in this simple special case. The conditional distribution P(S|Y=k) should be explicitly written out for completeness, because it is a crucial part of the test. So if I have understood correctly, the test will detect associations if the categories of Y differ in the relative variant frequency profiles (where the profile is over the markers). However, this kind of test is not sensitive if there are differences in the variant frequencies across the categories but not across the markers. Consider for example the setting with n=300, m=2 and numbers of observations in the cross-tabulation (X_1 and X_2 are the two markers):
Y=0,X_1=0: 100; Y=0,X_1=1: 50; Y=0,X_2=0: 100; Y=0,X_2=1: 50
Y=1,X_1=0: 150; Y=1,X_1=1: 0; Y=1,X_2=0: 150; Y=1,X_2=1: 0
It's clear that there is a difference between Y=0 and Y=1 in both markers, but P(S=1)=P(S=2)=P(S=1|Y=0)=P(S=1|Y=1)=P(S=2|Y=0)=P(S=2|Y=1)=1/2 and MI(S,Y) = 0, because there is no difference between X_1 and X_2.

After defining the test, the properties of the test in relation to finding which kind of associations are hardly discussed at all in the manuscript.

The paragraph on lines 147-152 on page 4 should rather be in the Notations and Existing Tests section.

The original study of the real-data example is not referenced at all (I'm assuming the data has been analysed previously). It is not entirely clear if the trait is categorical or could be modelled as ordinal. The real-data example has no "negative control", since all of the examined genes are assumed associated. Thus comparing different methods on it will mainly favour the one that is the least conservative. Since the top associations are bound to be known genes, discussing the functionality of some of them is not interesting in this manuscript as it does not contribute any evidence in favour of the proposed methods.

The simulations and the real-data example examine only the case with the number of categories being 3 (K=3). It is not clear how the method scales with K (while categorical trait is apparently the main novelty in the manuscript).

Experimental design

See Basic Reporting for all comments.

Validity of the findings

See Basic Reporting for all comments.

Additional comments

Overall, there are multiple concerns about the novelty, quality, and the results in this manuscript. See Basic Reporting for the specific comments.

---

## Round 0.2 · Minor Revisions

The manuscript has been substantially improved but there are a number of comments by the two reviewers that need to be addressed point by point.

Reviewer 1 ·

Basic reporting

I only have one inquiry:

Please add a column in Table 4 to show the number of SNPs in each gene regions. The methods like MFLM are designed for high-dimensional next-generation sequencing data (can be a combination of rare and common variants), and the information of # of variants in each gene region will make readers to know if the data are good for each method.

If the authors can add the information, the paper can be accepted.

Experimental design

no comments

Validity of the findings

no comments

Additional comments

I only have one inquiry:

Please add a column in Table 4 to show the number of SNPs in each gene regions. The methods like MFLM are designed for high-dimensional next-generation sequencing data (can be a combination of rare and common variants), and the information of # of variants in each gene region will make readers to know if the data are good for each method.

If the authors can add the information, the paper can be accepted.

Reviewer 2 ·

Basic reporting

The authors have addressed the main points of my previous review.

The writing is still not fluent in all places (judging based on my non-native language skills) and polishing it could make the argumentation and relationship to previous methods more clear, but I leave it to the editor's discretion if this would be required. The statistical method itself is presented clearly.

Experimental design

The statistic seems interesting and is compared to alternatives in extensive simulations.

There is still no negative control in the real-data example when comparing the proposed approach to alternative methods, so the current results only tell about the number of positive findings among all true (to the extent of current knowledge) positive. This is useful information, but it would be interesting to know how many genes the methods consider as significant outside of the 202 genes known to be associated to the trait. (But I would not necessarily require this for publication as the real truth about those genes is not anyway known.)

Validity of the findings

The rebuttal made me understand the statistic better and now the possible weaknesses are also mentioned more clearly in the text.

Additional comments

It would be great if the authors would make a software implementing the test available (at least in the future if not already upon publication).

---

## Round 0.3 · accepted · Accept

All the previous issues have been addressed and this manuscript can now be endorsed for publication on PeerJ.